# Replication cycle timing determines phage sensitivity to a cytidine deaminase toxin/antitoxin bacterial defense system

**Brian Y. Hsueh¤, Micah J. Ferrell, Ram Sanath-Kumar, Amber M. Bedore, Christopher M. Waters** ⓘ *

Department of Microbiology and Molecular Genetics, Michigan State University, East Lansing, Michigan, United States of America

¤ Current address: Meso Scale Diagnostics LLC, Rockville, Maryland, United States of America
* watersc3@msu.edu

**Data Availability Statement:** All data is accessible at: Waters, Christopher; Hsueh, Brian (2023). Replication cycle timing determines phage sensitivity to a cytidine deaminase toxin/antitoxin

## Abstract

Toxin-antitoxin (TA) systems are ubiquitous two-gene loci that bacteria use to regulate cellular processes such as phage defense. Here, we demonstrate the mechanism by which a novel type III TA system, *avcID*, is activated and confers resistance to phage infection. The toxin of the system (AvcD) is a deoxycytidylate deaminase that converts deoxycytidines (dC) to dexoyuridines (dU), while the RNA antitoxin (AvcI) inhibits AvcD activity. We have shown that AvcD deaminated dC nucleotides upon phage infection, but the molecular mechanism that activated AvcD was unknown. Here we show that the activation of AvcD arises from phage-induced inhibition of host transcription, leading to degradation of the labile AvcI. AvcD activation and nucleotide depletion not only decreases phage replication but also increases the formation of defective phage virions. Surprisingly, infection of phages such as T7 that are not inhibited by AvcID also lead to AvcI RNA antitoxin degradation and AvcD activation, suggesting that depletion of AvcI is not sufficient to confer protection against some phage. Rather, our results support that phage with a longer replication cycle like T5 are sensitive to AvcID-mediated protection while those with a shorter replication cycle like T7 are resistant.

## Author summary

Numerous diverse antiphage defense systems have been discovered in the past several years, but the mechanisms of how these systems are activated upon phage infection and why these systems protect against some phage but not others are poorly understood. The AvcID toxin-antitoxin phage defense system depletes nucleotides of the dC pool inside the host upon phage infection. We show that phage inhibition of host cell transcription activates this system by depleting the AvcI inhibitory sRNA, which inhibits production of phage and leads to the formation of defective virions. Additionally, we determined that speed of the phage replication cycle is a key factor that influences sensitivity to AvcID

bacterial defense system. figshare. Dataset. https://doi.org/10.6084/m9.figshare.22028480.v1.

**Funding:** This work was supported by National Institutes of Health (NIH) grants GM139537 and AI158433 to C.M.W. The funders had no role in study design, data collection and analysis, decision to publish, or preparation of the manuscript.

**Competing interests:** The authors have declared that no competing interests exist.

with faster replicating phage exhibiting resistance to its effects. This study has implications for understanding the factors that influence bacterial host/phage dynamics.

## Introduction

Bacteria must respond and adapt to a plethora of different challenges to survive and propagate. One such challenge is predatory bacteriophage (phage). To counter phage infection, bacteria have evolved a diverse repertoire of molecular phage defense mechanisms including Restriction/Modification (RMs), CRISPR/Cas, cyclic-oligonucleotide-based antiphage systems (CBASS), retrons, and toxin-antitoxin (TA) systems [1–7]. TA systems were first discovered on plasmids (*e.g.* Type I) and later were ubiquitously found on bacterial and phage chromosomes [8–10]. These modules typically constitute a two-gene operon that encodes diverse toxins along with a peptide or RNA antitoxin that neutralizes the toxin [11–13]. There are currently eight classes (I-VIII) of TA systems based on the nature of the antitoxin and the mechanism by which it regulates the toxin [11,13]. The toxins are generally proteins except for the type VIII system, in which the toxin is a small RNA (sRNA) [13,14]. In the case of type I, III, and VIII TA systems, the antitoxins are sRNAs while the rest of the classes have small peptide antitoxins [13]. Antitoxins are more abundant than their cognate toxins but are more labile, freeing the toxins to exert their growth-inhibition functions when expression of both genes is inhibited [15].

Though many past studies employed abiotic stressors (*i.e.* antibiotics, oxidative agents), to test the induction of type II TA systems, recent findings show that biotic stress, such as phage infection, can also activate TA systems [13,16]. RMs encoded in Type I TA systems inhibit phage infections and promote plasmid maintenance [2,8]. Similarly, other TA systems have been shown to limit phage infections [17–20]. Additionally, TA modules are not only clustered and closely connected to mobile genetic elements (MGEs), but they also mediate the stabilization of MGEs by limiting gene reduction [21]. TA modules also are highly abundant in free-living bacteria but not symbiotic, host-associated species [22], supporting their role as phage defense systems.

AvcID is a newly discovered, broadly conserved Type III TA system that encodes the AvcD toxin and AvcI antitoxin. AvcD is a deoxycytidine (dU) deaminase that deaminates dCTP and dCMP nucleotides to dUTP and dUMP, respectively, leading to a disruption in nucleotide metabolism after phage infection [23,24]. AvcI is a noncoding RNA that binds to and directly inhibits the activity of AvcD; however, the mechanism by which phages induce activity of the AvcID system remains unknown. Moreover, why AvcID inhibits infection of some phage but not others is not understood. Here, we demonstrate that upon phage infection, the AvcI sRNA antitoxin is rapidly lost, allowing AvcD to deaminate dC pools. This activation leads to less phage production and the production of defective phage virions. Contrary to our hypothesis, T7 phage resistant to AvcID-mediated protection still depleted AvcI and activated AvcD upon infection, suggesting other dynamics of phage/host interactions are important for sensitivity to AvcD. Our results suggest that the replication cycle of T7, or the time it takes from phage infection to cell lysis, is a key factor mediating sensitivity to AvcID as phage with rapid replication cycles are resistant to AvcID-mediated protection.

## Results

### AvcID provides phage defense in liquid cultures

Previous studies demonstrated that AvcID systems derived from *Vibrio cholerae, V. parahaemolyticus, and Escherichia coli* ETEC can reduce the dC nucleotide pool upon phage infection,

yet their respective resistance profiles as measured by efficiency of plaquing (EOP) assays are different [23,24]. For example, *V. parahaemolyticus* AvcID provided protection against T3, T5, T6, and SECΦ18 phages whereas *E. coli* ETEC AvcID provided protection against T3, SECΦ17, SECΦ18, and SECΦ27 [23]. To better understand the activation of AvcID systems and the molecular mechanisms underlying phage defense specificity, we studied the AvcID system derived from *V. parahaemolyticus* in a heterologous *E. coli* host as it provides robust protection against well-characterized T-type coliphages, such as T5. For the rest of this study, "AvcID" refers to the *V. parahaemolyticus avcID* we previously described [23].

Since the protection conferred by the AvcID system has thus far been based on comparing phage titers on agar plates [23], we wondered if the AvcID system could also confer protection in liquid culture as this would be a more tractable experimental system to explore the molecular mechanism of this phage defense. To answer this question, we generated *E. coli* cells harboring genes encoding for either the wild type AvcID (pAvcID) or the inactive AvcID$^{S49K +E376A}$ mutant enzyme (pAvcID*), expressed from their native promoters on a medium copy number plasmid. These *E. coli* strains were infected with either T5 or T7 at varying multiplicities of infection (MOI), and bacterial growth was tracked by measuring $OD_{600}$ over time. The $OD_{600}$ of T5 infected cultures harboring AvcID was higher throughout the experiment compared to cultures harboring AvcID* at all the MOIs tested, and at MOIs from 0.1–0.001 AvcID completely protected the culture from T5 (Fig 1A). Alternatively, AvcID showed no protection against T7 infection in liquid culture at any MOI tested (Fig 1B). These data show that AvcID provides *E. coli* with defense against T5 but not T7 in liquid culture.

## AvcD is activated by inhibition of host cell transcription

AvcI and AvcD form a complex in vitro, and AvcD is inhibited by the sRNA AvcI, suggesting AvcD inhibition is linked to the assembly of the complex [23]. A common mechanism that lytic phage employ upon infection is inhibition of host cell transcription [19,25–27]. Given that antitoxins are unstable, we hypothesized that AvcI is degraded upon inhibition of host cell transcription by phage, either through direct inhibition of transcriptional machinery or indirectly by host genome degradation, leading to activation of AvcD. To test this, we assessed the stability of AvcI by Northern blot from *E. coli* cells encoding the *avcID* locus on a plasmid under the expression of its native promoter. Likewise, to determine whether AvcD protein levels changed concurrently with changes in *avcI* RNA levels, we quantified the AvcD protein using a Western blot with antibodies specific to a C-terminal 6xHis tag. Notably, the full length

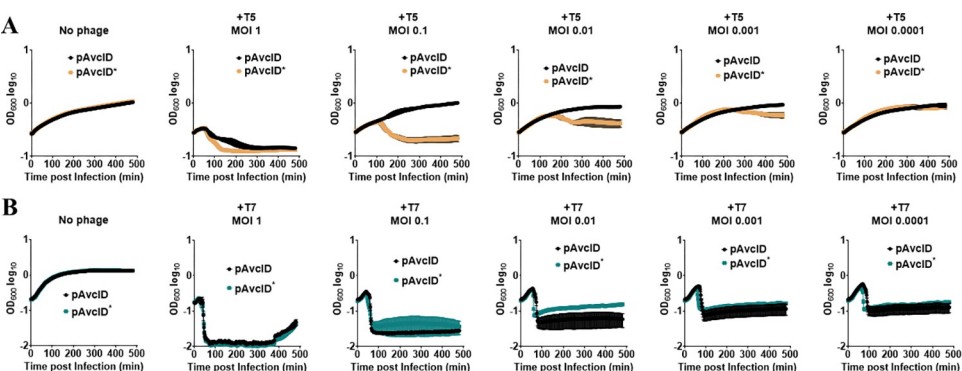

**Fig 1. The AvcID system provides phage defense in liquid culture.** Growth curves for *E. coli* with active (pAvcID) or inactive (pAvcID*) after infection with T5 (**A**) or T7 (**B**) phage at varying multiplicities of infection (MOI). Data represents the mean ± SEM of three biological replicate cultures.

of the *avcI* transcript is slightly smaller than 300 ribonucleotides, which is longer than the minimum functional length of AvcI of ~171 bases that was previously determined (S1 Fig) [23]. In some cases, a doublet of AvcI was observed for unknown reasons. As electrophoresis was performed in non-denaturing conditions, this doublet could be either due to alternate folding forms of AvcI or different RNA lengths, which is being investigated. Nevertheless, the levels of AvcI rapidly decreased when cells were treated with the transcriptional inhibitor rifampicin, showing that in the absence of new transcription, AvcI is degraded (Fig 2A). Importantly, spectinomycin, which inhibits protein synthesis instead of transcription, did not decrease *avcI* levels (Fig 2B), indicating that the degradation was specific to transcriptional shutoff. These results are consistent with our previous study showing that rifampicin but not spectinomycin activated AvcD in *V. cholerae* [23].

We also infected these cells with phages T5 and T7 and detected AvcI RNA and AvcD protein post infection (Fig 2C and 2D). The fact that AvcI is degraded faster during T7 compared with T5 is a surprising result given that AvcID provides protection against T5 phage but not T7. Concurrently, we also found that AvcD protein levels did not change significantly in any of the conditions tested (Fig 2A–2D), with the exception of T7 at the 20-minute timepoint. which is due to most cells being lysed by T7.

To determine whether AvcD is activated upon loss of AvcI, we measured the intracellular abundance of dCTP and dCMP using UPLC-MS/MS before and after infecting the cells containing active or inactive *avcID* with T5 or T7 phages. Surprisingly, both T5 and T7 infections significantly decreased intracellular dCTP and dCMP in cells containing active *avcID*, demonstrating that both phages activate AvcD (Fig 3A and 3B). We also noted that T5 decreases intracellular dCMP in cells containing inactive *avcID* (Fig 3B). We hypothesize this result is due to a T5 encoded 5' monophosphatase (*dmp*) which participates in the final stages of host DNA degradation by dephosphorylating 5'-dNMP's substrates [28]. Collectively, these results suggest that inhibition of transcription coupled with the instability of the *avcI* RNA leads to the release of existing AvcD from inhibition upon both T5 and T7 phage infection, but activation of AvcD is not sufficient to protect against T7 phage infection.

## AvcID drives production of defective T5 phage

We found that AvcID provides resistance to T5 but not T7 (Fig 1A and 1B), yet both phages induce the degradation of AvcI and the activation of AvcD deamination (Figs 2 and 3). To

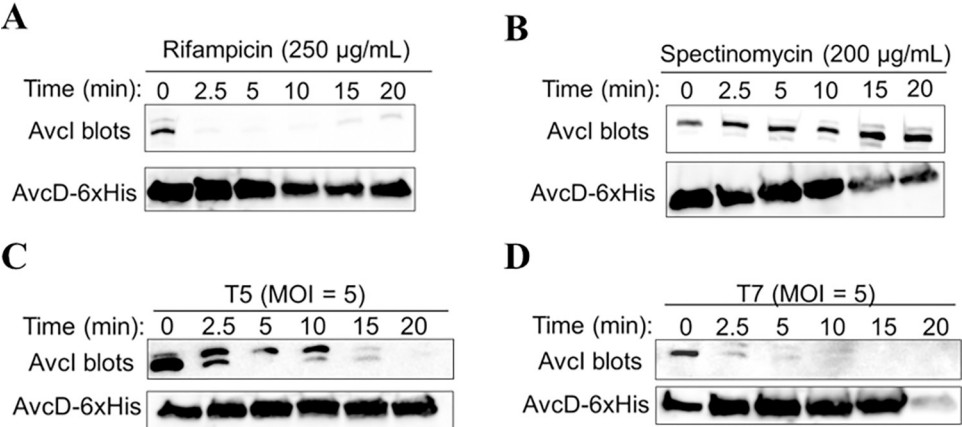

**Fig 2. Transcriptional shutoff leads to the degradation of *avcI*.** Shown are Northern blots of *avcI* RNA using a biotinylated probe complementary to *avcI* (top) and Western blots of AvcD-6xHis using anti-6xHis antibody (bottom) during rifampicin treatment (250 μg/mL) (**A**), spectinomycin treatment (200 μg/mL) (**B**), T5 infection (**C**), and T7 infection (**D**) at a MOI of 5.

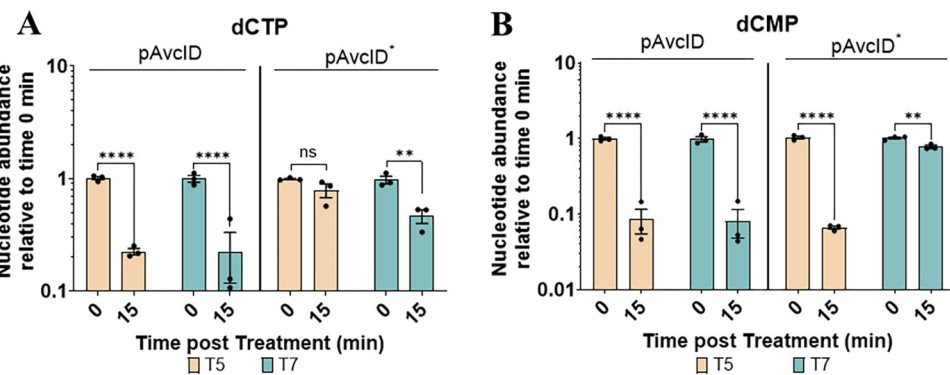

**Fig 3. AvcD is activated by T5 and T7 phages.** In vivo abundance of dCTP (**A**) and dCMP (**B**) in an *E. coli* host carrying pAvcID or pAvcID* with its native promoter before and after infection of T5 (MOI = 5) or T7 phage (MOI = 5). Nucleotides were measured using UPLC-MS/MS and normalized to total protein. Data represents the mean ± SEM of three biological replicate cultures, Two-way ANOVA with Dunnett's post-hoc test, and ns indicates not significant.

further understand this difference, we quantified the production of phage and the viability of *E. coli* hosts during liquid T5 (MOI 0.1) and T7 (MOI 0.01) phage infection assays for *E. coli* carrying either AvcID or inactive AvcID*. Infected *E. coli* cells were separated from the phage lysates by centrifugation to measure colony forming units (CFUs) and plaque forming units (PFUs). A delay in population collapse was observed for cells harboring AvcID infected with T5 compared to cells harboring inactive AvcID. (Fig 4A). Furthermore, the AvcID-containing population generated ~100-fold fewer PFUs than AvcID*-containing cells (Fig 4B), supporting the notion that AvcID inhibits the accumulation of functional T5 phages. Consistent with our liquid infection results described above, AvcID did not impact the number of CFUs and PFUs when cells were infected with T7 (Fig 4D and 4E).

Since a plaque assay only quantifies viable phages, we speculated that the total viral particles produced could be underestimated if some of those virions were non-viable. To determine

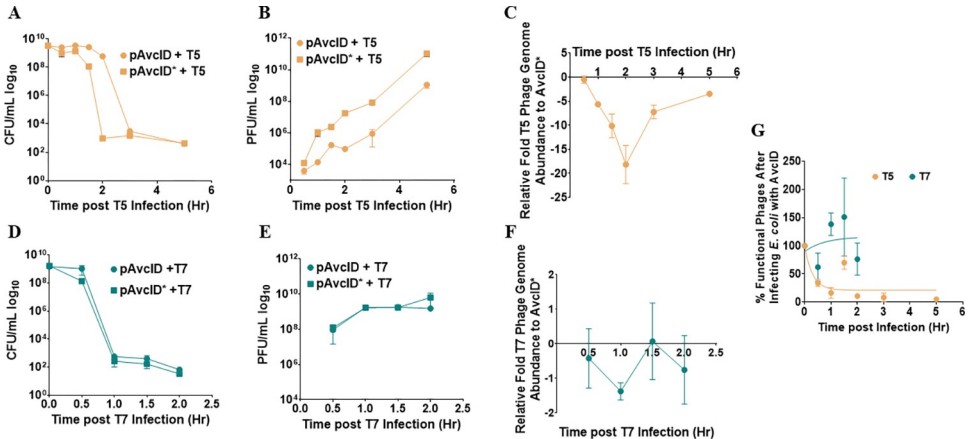

**Fig 4. AvcID reduces the functionality of T5 but not T7 phage.** Survival of *E. coli* encoding the indicated AvcID systems as measured by CFU after infection with T5 (MOI 0.1) (**A**) or T7 (MOI 0.01) (**D**). PFU quantification over time in cultures of pAvcID or pAvcID* containing cells infected with T5 (**B**) or T7 (**E**). Relative T5 (**C**) or T7 (**F**) genome abundance comparing *E. coli* expressing pAvcID or inactive AvcID* over time. Percent viable phage after infecting cells containing AvcID with T5 or T7 phages (**G**). Data represents the mean ± SEM of three biological replicate cultures.

total virions produced, we quantified the abundance of a specific phage gene for T5 and T7 in the phage particle samples using qPCR. Importantly, these samples had been treated with DNase ensuring that only genomes protected by phage virions were quantified by this assay. Our results indicated that the total number of T5 phage genomes decreased over time in infected cultures of AvcID-containing cells compared to AvcID*-containing cells. However, the magnitude of this decrease was less than that observed for the difference in PFUs between the two samples (Fig 4B and 4C). For example, at the two-hour time point, there was a difference of ~100-fold in PFUs but only a difference of 20-fold in virions measured with qPCR. We observed no significant difference in genome abundance between AvcID and AvcID* encoding cultures when infected with T7 (Fig 4F).

The greater magnitude difference for T5 of PFUs compared to phage genomes suggested that the majority of virions produced from AvcID-containing cells contained genomes that were defective to infect new cells and form plaques. We calculated the percentage of viable phages by quantifying the ratio between PFUs and genome abundance of AvcID or AvcID* infected cultures. Using this analysis, we estimated that by 30 min only 30% of T5 phage derived from cells containing AvcID were functional, and the proportion of functional phage generally decreased over time, suggesting that AvcID drives formation of defective T5 phage virions (Fig 4G). In contrast, a similar analysis for T7 indicates that nearly all the T7 phages were viable even when they were from cells encoding *avcID* (Fig 4G). This indicates that AvcID confers protection by both decreasing phage replication and increasing defective phage production for T5 while T7 can overcome these negative effects of AvcID through an unknown mechanism.

Consistent with our observation that defective T5 phage are generated from *avcID*-encoding cells, transmission electron microscopy (TEM) images of negatively stained samples revealed that particles produced from cells containing active AvcID have more defective phage capsids—*i.e.* broken particles, or capsids with aberrant morphology, containing no genome and without attached tails (Fig 5A), compared to T5 phage from cells containing AvcID* (Figs 5B and S2). Images of negatively stained particles of T7 isolated from *avcID* encoding cells showed no such defects and the virions looked normal (Fig 5C). Together, the TEM results corroborate our previous experiments confirming that the AvcID system inhibits production of functional T5 phage particles.

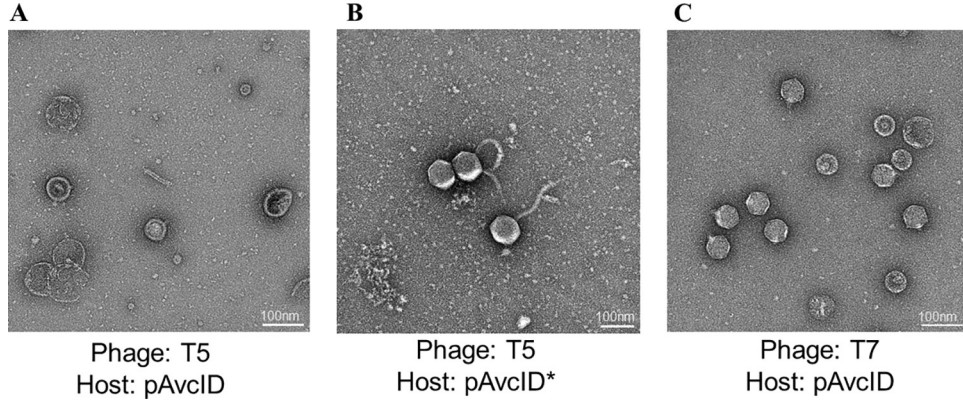

Phage: T5
Host: pAvcID

Phage: T5
Host: pAvcID*

Phage: T7
Host: pAvcID

**Fig 5. TEM of AvcID-induced Phage Defense.** Transmission Electron Microscopy (TEM) of T5 (**A, B**) or T7 (**C**) from *E. coli* host carrying pAvcID (**A**, **C**) or pAvcID* (**B**). Samples were negative stained with 1% (w/v) uranyl acetate. Scale bar 100 nm. All samples were analyzed in three biological replicates with similar results.

## Ung and Dut do not contribute to AvcID antiphage defense

We have previously demonstrated that AvcD can deaminate dCTP to dUTP. Increased dUTP concentrations in cells may lead to an increased frequency of dUMP being incorporated into the phage genomic DNA in place of dTMP by DNA polymerases during replication. Incorporated dUMP in genomic DNA can be targeted by the DNA repair enzyme uracil-N-glycosylate (Ung), leading to formation of an abasic site that could block DNA replication. Moreover, an apurinic or apyrimidinic endonuclease can then cleave the DNA at the abasic site, resulting in a nicked DNA strand [29,30]. We hypothesized that the decreased viability of phage produced in AvcID expressing cells could be due to incorporation of uracils in the genomic DNA of T5 while T7 was resistant to such activity. The uracils in the genomic DNA could be subjected to the excessive repair by Ung upon infection of the uninfected bacterial hosts.

To determine whether AvcID and Ung function together to reduce phage infection, we infected *E. coli* MG1655 or Δ*ung E. coli* NR8052 encoding *avcID* or *avcID** with T5 phage, measured the relative phage titer, and further tracked bacterial growth by $OD_{600}$ over time. We hypothesized that the Δ*ung* mutant would not exhibit as robust of protection from T5 as the WT *E. coli*. Contrary to our hypothesis, the $OD_{600}$ of both strain backgrounds carrying active AvcID exhibited similar protection, suggesting that Ung is not required for AvcID to protect *E. coli* from T5 phage (Fig 6A). When comparing relative phage titer, we observed no difference in AvcID-mediated protection from T5 in the presence or absence of *ung* (Fig 6B). Finally, we infected *E. coli* MG1655 containing either AvcID or AvcID* with T5 phage while overexpressing a dUTPase (*dut*) to reduce accumulation of dUTP, thereby preventing potential incorporation of uracil into phage genomes. The overexpression of Dut had no effect on

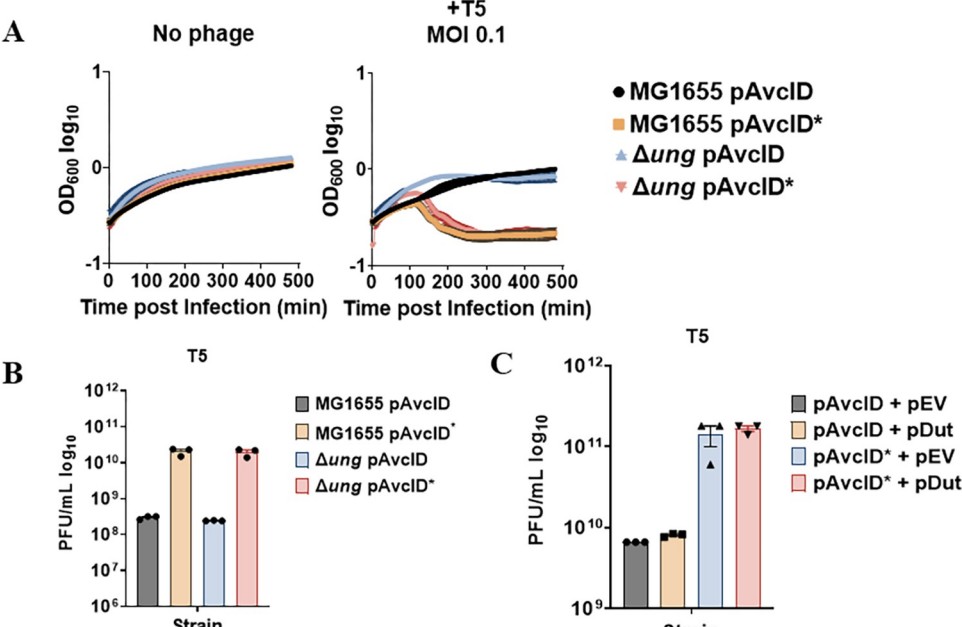

**Fig 6. Ung and AvcID do not function together to provide phage protection.** (**A**) Growth curves for *E. coli* MG1655 or Δ*ung* mutant with pAvcID or pAvcID* after infection with T5 phage at varying multiplicities of infection (MOI). Data represents the mean ± SEM of three biological replicate cultures. (**B**) Measurement of phage titer on WT *E. coli* MG1655 or Δ*ung E. coli* encoding either active or inactive AvcID system infected with T5 phage. Data represents the mean ± SEM of three biological replicate cultures. (**C**) Measurement of phage titer on WT *E. coli* MG1655 co-expressing Dut and either active or inactive AvcID system infected by T5 phage. Data represents the mean ± SEM of three biological replicate cultures.

the phage defense conferred by AvcID (Fig 6C). Together, these data suggest that accumulation of dUTP or incorporation of uracils into the phage genome does not contribute to AvcID-mediated protection.

## T7 mutants with a slower replication cycle are susceptible to AvcID

Our results demonstrated that both T5 and T7 activate AvcID, but this system only protects against T5 infection. When considering this discrepancy, we noted that T7 has a much faster replication cycle compared to T5 (~22.5 min vs. 60 min), suggesting a model in which T7 could replicate enough genomes before AvcID is activated to mitigate its protective effects. Consistent with this model, quantification of T7 genome accumulation using Q-PCR showed that most genome replication had occurred by 10 minutes and there was no significant difference in the number of replicated phage genomes in the presence of AvcID (S3 Fig). Alternatively, we had previously demonstrated using the same approach that AvcID significantly reduced T5 genome replication [23].

To test if the speed of the replication cycle is a key factor in AvcID-sensitivity, we treated T7 phage *in vitro* with the alkylating agent methyl methanesulfonate (MMS). Previous research demonstrated that this treatment increases the length of the first replication cycle of phage infection by creating lesions in the DNA that must first be repaired by the host DNA repair machinery before replication can be initiated [31]. Therefore, MMS treated phage would have a delayed replication cycle during the first infection cycle as the DNA is repaired, but subsequent cycles would resemble untreated T7. If our model is correct, we expect to observe initial AvcID-mediated protection against MMS treated T7 that would be lost as the phage DNA is repaired over time.

When infecting the cells at a MOI of 0.1, AvcID showed minimal protection against untreated T7 and MMS-treated T7 (S3A and S3B Fig). However, at a MOI of 0.0001, AvcID delayed the complete lysis of the population by MMS-treated T7 phage (S3C and S3D Fig). Ultimately, however, the rate of the population drop was identical in all conditions. We interpret this result to mean that AvcID was able to protect a significant portion of cells from MMS-treated T7 during the initial round of replication, but ultimately as the genomes of the phage were repaired subsequent replication of undamaged T7 overcame the AvcID-encoding bacterial population. Such a population dynamic is consistent with the delayed replication cycle of MMS treated T7 rendering the phage sensitive to AvcID.

To further explore the impact of replication cycle time on AvcID-mediated protection, we infected *E. coli* carrying AvcID or AvcID* with the T7 mutant phage T7^412 (gift of Ian Molineux) that has a delayed replication cycle of 40 min. T7^412 encodes deletions of genes 0.5–0.7 with the T7 RNA polymerase gene, gene 1, integrated near gene 12. Unlike WT T7, the plaque formation of the T7^412 mutant phage was completely inhibited by *avcID* (Fig 7A and 7B). We next quantified the viability of hosts, production of functional phage, and total phage genome abundance with qPCR of T7^412 in AvcID and AvcID* encoding cells. Cells containing AvcID infected with T7^412 had more viable CFUs than cells carrying AvcID* (Fig 7C), and the AvcID-containing cells had up to ~5 orders of magnitude less T7^412 PFUs than AvcID*-containing cells (Fig 7D). Additionally, the total number of T7^412 genomes quantified using qPCR decreased over time in infected cultures of AvcID-containing cells compared to the inactive variant (Fig 7E). Using these results, we estimated the proportion of functional T7^412 by 30 min was only approximately 2% of the total phage virions (Fig 7F). These data indicate that AvcID confers protection against T7^412 by increasing defective phage production and generating non-functional phage, similar to our results for T5.

To determine if the loss of gene 0.5–0.7 in phage T7^412 contributed to AvcID-mediated sensitivity, we performed two additional experiments. T7^C74 is a mutant T7 phage that has the

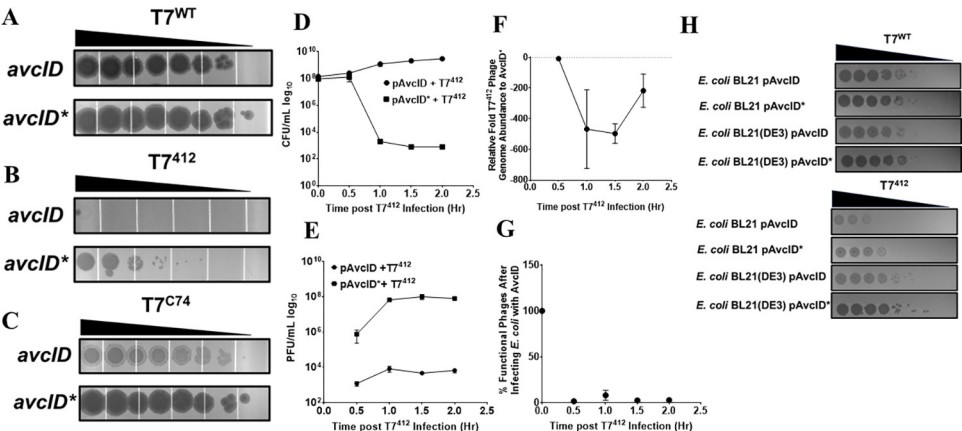

**Fig 7. AvcID reduces the functionality of T7^412 mutant phage.** Representative images of tenfold serial dilution plaque assays of T7^WT (A), T7^412 (B), or T7^C74 (C) phages spotted on *E. coli* MG1655 expressing either active (top) or inactive (bottom) *avcID* system. Images are representative of three replicates. (D) CFU quantification of *E. coli* MG1655 over time in cultures indicated AvcID systems after infection with mutant T7^412. (E) PFU quantification over time in cultures of indicated AvcID systems-containing cells infected with T7^412. (F) Relative T7^412 genome abundance comparing *E. coli* expressing pAvcID or inactive AvcID* over time. (G) Percent viable phage after infecting cells containing AvcID with T7^412. Data represents the mean ± SEM of three biological replicate cultures. (H) Representative images of tenfold serial dilution plaque assays of T7^WT (top) or T7^412 (bottom) phages spotted on *E. coli* BL21 or BL21(D3E) expressing either active or inactive *avcID* system. Images are representative of three replicates.

genes 0.5–0.7 deleted but gene 1 is encoded in its native location. T7^C74 formed equivalent plaques regardless of AvcID similar to WT T7, although we noted that the plaques formed on AvcID-expressing cells were more cloudy than those on AvcID* containing cells (Fig 7C). Liquid growth assays demonstrated that T7^C74 is resistant to AvcID at all MOIs tested whereas T7^412 is sensitive to AvcID at MOIs 0.1–0.0001 (S5 Fig). Importantly, the replication cycle timing of T7^C74 was similar to T7 WT (S5 Fig). In addition, infection of T7^412 was completely resistant to AvcID-mediated defense in a Bl21(DE3) host that restored early production of T7 RNA polymerase (Fig 7H). These results demonstrate that it is solely the defect in RNA polymerase production and not any other mutations in the phage that confer sensitivity of T7^412 to AvcID.

## Discussion

Phage predation is a constant evolutionary pressure that shapes the diversity and fitness of bacteria driving the evolution of multiple antiphage defense systems. The underlying mechanisms of certain antiphage defenses such as RMs, which utilize DNA modifications to distinguish host and foreign DNA, are well-characterized [2,32,33]. On the contrary, the mechanism of activation of the cyclic nucleotide-based systems (*i.e.*, CBASS) in response to phage infection is generally not understood [3,34] although two recent studies show that depletion of intracellular folates activates CBASS [35,36]. Moreover, although many novel phage defense systems have been recently identified, it is typically unclear why they provide protection against some phage but not others [7,24,37,38]. Here, we reveal the mechanism of how the AvcID TA system is activated in response to phage infection and its impact on the phage's morphogenetic pathway. We also determined that the speed of phage replication cycle is an important factor that drives AvcID to specifically protect against longer replicating phage like T5 while it is ineffective against faster replicating phage like T7.

Cessation of transcription, either specifically by inhibition of transcription machinery or non-specifically by host genome degradation, is a hallmark of infection by many phages

[17,27,28,39]. Type 1 TA Hok/Sok, one of the earliest studied TA systems, also did not protect against T7 [17]. Our results demonstrate that transcriptional shutoff leads to the degradation of AvcI, releasing AvcD to deaminate dC nucleotides. This mode of activation is consistent with other TA systems, such as the ToxIN system [19,40] and previous work showing that inhibition of transcription activates AvcD, although the mechanism for this activation was not known [23,24]. This work is the first to show that this activation is due to degradation of the AvcI sRNA antitoxin. The RNA transcripts of AvcI antitoxin are produced at high abundance compared to AvcD transcripts [23]. The two bands of AvcI transcripts in northern blot have been shown with other RNA antitoxins [40], and they could represent alternate folding structure of AvcI as electrophoresis was performed on a non-denaturing gel. In other TA systems, typically a Rho-independent terminator located between the toxin and antitoxin genes [41,42]. This may partially explain the multiple bands of AvcI RNA transcripts as the longer transcript is the precursor to the smaller, processed transcript. However, sequence analysis did not predict such a terminator between *avcI* and *avcD*. Thus, how the AvcI sRNA is formed and produced at much higher levels than AvcD and the active form of AvcI sRNA required to inhibit AvcD require further study.

Though AvcID did not provide protection against T7, *avcI* transcripts were rapidly degraded upon T7 infection, implying that *avcI* levels decrease when cells are infected by the phage either through direct inhibition of transcription or indirect inhibition such as degradation of the host genome. However, the activation of AvcD does not have any detrimental effect on the viability of T7 (Fig 4), in contrast to T5, nor replication of T7 genomes (S3 Fig), implying that T7 has evolved to disregard any detrimental effects inflicted by AvcID or similar phage defense systems. Given that we observed no difference in viable phage from functional AvcID versus AvcID* containing cells (Fig 4), we suggest that even in the presence of AvcID and depleted dCTP, T7 replicates enough genomes (S3 Fig) to fully package all the capsid heads that are produced. Phage can synthesize more genomes than capsid heads, consistent with this interpretation [43,44]. It should be noted that examination of AvcD orthologs from *E. coli* expressed in a heterologous *E. coli* host did show protection against T7 using a plaquing assay [24]. This result highlights the high degree of specificity in phage protection for each defense system. The reason for this difference between these studies is not clear, but we speculate it may be due to the specific molecular features of the two AvcD systems being studied or differences in the T7 phage that were tested.

Our analysis of population dynamics of T5 infected cultures suggested that AvcID not only impacts overall synthesis of T5 genomes, but it also leads to the formation of non-viable phage capsids. This conclusion is supported by TEM images that reveal most of the T5 phages are defective when AvcD is active. The depletion of dCTP and dCMP could have downstream impacts on the timing of DNA packaging or stability of phage virions, thus accounting for the reduced number of functional phages. The mechanism by which AvcID generates non-viable phage capsids is under investigation.

The growth of several well-known phages is inhibited when their DNA contains dUMP and Ung is present in the host cells. For example, to counter this negative effect T5 encodes its own dUTPase for reducing the dUTP level such that dUMP is limited in its genome [45]. However, the presence of the AvcID system prevents this T5 infection even in the absence of Ung, indicating that dUMP incorporation into the phage genome may not be the cause for the phage viability defect. Moreover, the T5 protein, T5.015, interacts with Ung, which is thought to be a mechanism to overcome Ung-mediated restriction [45]. It is possible that a T5 defective in this protein would exhibit more sensitivity to AvcID-mediated protection. Rather, our evidence suggests it is the depletion of the dC pool that is responsible for the reduction in functionality of T5 phage particles. Remarkably, the depletion in the dC pool has no effect on T7 viability even though its G/C content

is approximately 52% [46]. T7 phage degrade the host genome and incorporate it into its own genome [47], which may partly offset the decreased dCTP nucleotide pool.

We obtained two lines of evidence that linked phage replication cycle to AvcID sensitivity. First, treatment of phage virions with MMS, which is known to delay the replication cycle time [31], enhanced sensitivity to AvcID, but this defense was only transient (S4 Fig). As T7 phage were repaired and replicated, the bacterial culture ultimately succumbed to phage infection regardless of AvcID. Secondly, the resistance of T7 to AvcID can be completely inhibited by altering its replication cycle, as T7$^{412}$ had a slower replication cycle and was completely susceptible to AvcID, even though this phage was fully capable of infecting and eradicating a non-AvcID expressing population [48]. This phage has a deletion of genes 0.5–1 with gene 1, the T7 RNA polymerase, inserted downstream of gene 12. The net result of these mutations is a delay in the synthesis of phage proteins and genome replication, increasing the duration of the replication cycle. The sensitivity of T7$^{412}$ suggests that the rapid replication cycle of T7$^{WT}$ enables it to outrun the AvcID system to produce new virions before AvcID has completely depleted dC pools. A third line of evidence supporting the importance of the replication cycle speed in resistance to AvcID was illustrated in a recent study of a *V. cholerae* ICP3 phage, named M1Φ, that was partially restricted by the native *avcID* locus [49]. In this study, an AvcID-resistant mutant M2Φ was isolated, and strikingly the replication cycle of this mutant (20 min) was twice as fast as the original phage (40 min) [49]. Importantly, this phage had only one mutation in the entire genome in the phage DNA polymerase, and that change alone was sufficient to counteract AvcID. This result studying *avcID* in its native genome context and host is consistent with our conclusion that replication cycle timing is an important driver of AvcID sensitivity and resistance [49].

Prior work on Type III TA systems suggests they are associated with abortive infections (Abi), which is defined as the host committing altruistic suicide to prevent phage replication [40]. However, overexpression of AvcD does not lead to cell death but does impair genome replication, and this effect can be rescued by inhibiting expression of the toxin or overexpressing *avcI* in trans [23]. Given that *avcI* is degraded, subsequently releasing existing AvcD to deaminate dC pools upon phage infection, we propose that protection conferred by AvcD is not through abortive infection. This conclusion is supported by our observation that infection of AvcID containing cells with a high MOI does not enhance killing of the host cells (Fig 1), and overexpression of AvcD does not lead to cell death [23].

Similar to the AvcID system, bacterial dGTPases protect against phage infection by dephosphorylating dGTP to dG to inhibit phage DNA replication and that this system is also activated upon phage-induced transcriptional shutoff [24]. It is, however, unclear whether the dGTPase system is a TA system. While other types of TA systems have been demonstrated to have antiphage properties, whether they are activated in a similar mechanism as the Type III systems is unclear. Recently, the DarTG type II TA system was shown to provide phage defense by ADP-ribosylating phage DNA to disrupt DNA replication [38]. ParST, another type II TA system, exerts its effect via modification of cellular target Prs, which is involved in nucleotide biosynthesis, though the ParST system has not been demonstrated to be involved in phage defense. The mechanism of AvcID bears a resemblance to both DarTG and ParST but is distinct from both in terms of the mechanism for toxin function and activation. This suggests that manipulating nucleotide pools is a conserved function of many TA systems and antiphage defense mechanisms.

## Materials and methods

### Bacterial strains, plasmids, and growth conditions

The strains, plasmids, and primers used in this study are listed in S1–S3 Tables. Unless otherwise stated, cultures were grown in Luria-Bertani (LB) at 37˚C and supplemented with

ampicillin (100 μg/mL), kanamycin (100 μg/mL), and isopropyl-β-D-thiogalactoside (IPTG) (100 μg/mL) when needed. *E. coli* BW29427, a diaminopimelic acid (DAP) auxotroph, was additionally supplemented with 300 μg/mL DAP. Plasmids were introduced into *E. coli* MG1655 or *E. coli* NR8052 through biparental conjugation using an *E. coli* BW29427 donor. P$_{tac}$ inducible expression vectors were constructed by Gibson Assembly with inserts amplified by PCR and pEVS143 [50] each linearized by EcoRI and BamHI. Transformation of *E. coli* for ectopic expression experiments was performed using electroporation with DH10b for expression of pEVS143 derived plasmids.

## Phage propagation

Coliphages were propagated on *E. coli* MG1655 grown in LB, and their titer was determined using the small drop plaque assay method, as previously described [3,51]. Briefly, 1 mL of overnight cultures were mixed with 50 mL of MMB agar (LB + 0.1 mM MnCl2 + 5 mM MgCl2 + 5 mM CaCl2 + 0.5% agar), tenfold serial dilutions of phages in MMB were spotted (5 μL) and incubated overnight at room temperature. The viral titer is expressed as plaque forming units per mL (pfu/mL).

## Phage infection in liquid culture

Overnight cultures of *E. coli* carrying the indicated AvcID plasmids were subcultured and grown to an OD$_{600}$ of 0.3 and then mixed with phage at the indicated MOIs. A 150 μL aliquot of the mixtures were put into 96-well plates, and growth was measured at 2.5 min intervals with orbital shaking on a plate reader (BioTek H1 (T5 only) or a SpectraMax M6 (rest of the phages)) at 37˚C for 8 hours. Data represents the mean ± SEM, *n* = 3.

## Plaque assays and imaging

*E. coli* MG1655 cells with indicated plasmids were grown in LB with 100 μg/mL ampicillin overnight at 37˚C. Overnight cultures are subcultured 1:500 in melted MMB agar and solidified at room temperature. Overnight cultures of *E. coli* MG1655 with inducible plasmids (pEV or pDut) were subcultured 1:1000 in LB with 100 μM IPTG and grown until an OD$_{600}$ of 1.0. The cultures were then subcultured 1:500 in melted MMG agar supplemented with 100 μM IPTG and let to solidify at room temperature.

   *E. coli* BL21 and BL21(DE3) transformed with pBYH67 or pBYH83 were grown overnight in MMB+ 100 μg/mL ampicillin. Strains were subcultured 1:100 into pre-warmed MMB broth with 100 μg/mL ampicillin and 100 μM IPTG. Cultures were grown at 37˚C for 4 hours and diluted 1:100 into tempered molten MMB agar supplemented with 100 μg/mL ampicillin and 100 μM IPTG, poured into plates and allowed to solidify.

   Tenfold serial dilutions of coliphages in MMB medium were spotted (5 μL) and incubated overnight at room temperature. We note that in this assay, protection of AvcID was temperature dependent and lost at 37˚C. This temperature dependency is currently under investigation. The images of the plaques were taken using ProteinSimple AlphaImager HP system.

## RNA extraction for Northern blot following phage infection

RNA isolation and qRT-PCR analysis were carried out as previously described [52]. Briefly, triplicate overnight cultures of *E. coli* carrying pAvcI-AvcD-6xHis were subcultured 1:100 in LB and grown to an OD$_{600}$ of 0.3. 1 mL of each replicate was pelleted and flash-frozen by the ethanol-dry ice slurry method. RNA was extracted using TRIzol reagent following the

manufacturer's directions (Thermo Fischer Scientific). RNA quality and quantity were determined using a NanoDrop spectrophotometer (Thermo Fischer Scientific).

## RNA probe synthesis and purification

The method for RNA probe production was modified from a previously described protocol [23]. The AvcI DNA template for *in vitro* transcription was PCR amplified from pAvcI using Q5 High-Fidelity DNA Polymerase (NEB). To incorporate the T7 promoter into the final AvcI DNA template, the forward primer included the T7 promoter sequence prior to the homologous sequence for AvcI. Additionally, the first two residues of the reverse primer were 2'-OMe modified to reduce 3'-end heterogeneity of the transcript [53]. The PCR reaction was analyzed using a 1% agarose gel, and the band corresponding to the AvcI DNA template was excised and gel purified using Promega Gel Extraction and PCR clean up kit. AvcI RNA was synthesized by *in vitro* transcription using the T7-AvcI reverse complement DNA template and the HiScribe T7 High Yield RNA Synthesis Kit (NEB). Bio-11-UTP was included during the transcription reaction for Northern blot detection purposes. The transcription reactions were incubated at 37°C for 4 h. Following transcription, DNase I (NEB) was added to a final concentration of 1X per reaction and incubated at 37°C for an additional 15 min. AvcI was then purified using Monarch RNA Cleanup Kit (NEB). Purity of the product was evaluated using a 1.0% TBE agarose gel. Individual aliquots of AvcI were flash-frozen using liquid nitrogen and stored long-term at -80°C.

## Northern blot analysis following phage infection and half-life quantification and analysis

1.5–2 μg total RNA was diluted 1:1 in 2x sample buffer (Invitrogen), loaded onto 7.5% TBE-Urea PAGE gels, along with biotinylated sRNA ladder (Kerafast),and ran for 30 min or until the front dye reached ~1 cm above the bottom of the gel at 200 V. RNA was then transferred to BrightStar-Plus Positively Charged Nylon Membrane (Invitrogen) with a Fisherbrand Semidry Blotting Apparatus (Fisher Scientific) and ran for 1 h at 250 mA. RNA was then crosslinked to the membrane using the CX-2000 crosslinker compartment of the UVP HybriLinker HL-2000 (Fisher Scientific). Each side of the membrane was crosslinked at 1200 μJ twice and dried at 50°C for at least 30 minutes to improve sensitivity. The membranes were pre-hybridized for at least 60 minutes at 60°C in ULTRAhyb Ultrasensitive Hybridization Buffer (Invitrogen) with gentle shaking. Next, the pre-hybridization buffer was removed, and hybridization buffer containing 1 nM of purified probe was added. The membrane was hybridized for 12–16 hours at 60°C with gentle shaking. Next, the membrane was rinsed twice every five minutes with 2x saline-sodium citrate (SSC) buffer, 0.1% SDS at 60°C and then twice every 15 minutes with 0.1X SSC, 0.1% SDS at 60°C. The biotin-labeled probes were detected using a Chemiluminescent Nucleic Acid Detection Module (Thermo Scientific) at RT. The membranes were then imaged using the Amersham Imager 600. To determine the half-life of *avcI*, the band intensities were analyzed using the Fiji software and normalized to respective 0 min band intensity [54]. All Northern blots shown are representative of two independent biological replicates.

## Western blot analysis of AvcD

Cells were collected in the same method as RNA extraction. Pellets were then resuspended at $OD_{600}$ = 15 (~20 μL) in 2x Laemmi loading dye supplemented with 10% β-mercaptoethanol v/ v, denatured for 10 min at 95°C, and centrifuged at 15k x g for 10 min. Samples were then analyzed by 4–20% SDS-PAGE gels (Mini-PROTEAN TGX Precast Protein Gels, Bio-Rad) alongside size standards (Precision Protein Plus, Bio-Rad or PageRuler Plus Prestained Protein

Ladder, Thermo Scientific). Gels were run at room temperature for 60 min at 120 V in 1x Tris/glycine/SDS running buffer. Proteins were transferred to nitrocellulose membranes (Optitran). The membranes were blocked using 5% skim milk and incubated with 1:5000 THE His Tag Antibody, mAb, Mouse (GenScript) followed by 1:4000 Goat Anti-Mouse IgG Antibody (H&L) [HRP], pAb (GenScript), treated with Pierce ECL Western Blotting Substrate, and imaged using an Amersham Imager 600. Western blots shown are representative of two independent biological replicates.

## CFU/PFU measurements pre- and post-phage infection

Overnight cultures were subcultured and split into two 10 mL aliquots and grown to an $OD_{600}$ of ~0.3. One aliquot was mixed with phage (T5, MOI = 0.1; T7, MOI = 0.01; $T7^{412}$, MOI = 0.01) and the other with an equal volume of LB (uninfected control). Both were grown in a shaking incubator (210 rpm) at 37°C. At each indicated timepoint, 1.5 mL of culture was spun down. The supernatant from each tube was filter sterilized with 0.22 µM filter and transferred to a new tube, and the bacterial cell pellets were washed twice with equal volume of LB to remove unadsorbed phage. For PFU measurements, the supernatants were serially diluted in MMB medium (LB + 0.1 mM MnCl2 + 5 mM MgCl2 + 5 mM CaCl2) and 5 µL of each dilution was spotted on a lawn of bacteria seeded in MMB agar plate (MMB + 0.5% agar). PFU plates were then grown at RT overnight and plaques quantified the following day. For CFU measurements, resuspended cell pellets were then incubated at 37°C for 5–10 minutes before being serially diluted 10-fold in PBS and 5 µL of each dilution was spotted on LB plates. CFU plates were then grown at 37°C overnight and colonies were quantified the following day.

## UPLC-MS/MS dNTPS quantification

Deoxynucleotide concentrations were determined as previously described [23] with minor modifications. Briefly, to measure the nucleotides after phage infection, cells were grown in LB overnight at 37°C. Overnight cultures were subcultured 1:100 in LB and grown to an $OD_{600}$ of ~0.3. 3 mL of culture were collected for a time zero reading: 1.5 mL for dNTPs quantification and 1.5 mL for total protein quantification. The cultures were then infected with phage (T7, MOI of 5), and an additional 3 mL were removed at each indicated subsequent time point. Culture aliquots were collected by centrifugation at 15k x g for 1 min. Pellets were resuspended in 200 µL of chilled extraction buffer [acetonitrile, methanol, ultra-pure water, formic acid (2:2:1:0.02, v/v/v/v)]. To normalize in vivo nucleotide samples, the other 1.5 mL aliquot pellet was centrifuged at 15,000 x g for 1 min, resuspended in 200 µL lysis buffer (20 mM Tris·HCl, 1% SDS, pH 6.8), and denatured for 10 min at 95°C. Denatured lysates were centrifuged at 15,000 x g for 1 min to pellet cellular debris, and the supernatant was used to quantify the total protein concentration in the sample by using the DC protein assay (Bio-Rad) and a BSA standard curve [55]. The concentrations of deoxynucleotides detected by UPLC-MS/MS were then normalized to total protein in each sample.

All samples resuspended in extraction buffer were immediately incubated at -20°C for 30 min after collection and centrifuged at 15,000 x g for 1 min. The supernatant was transferred to a new tube, dried overnight in a speed vacuum, and finally resuspended in 100 µL ultra-pure water. Experimental samples and deoxynucleotides standards [1.9, 3.9, 7.8, 15.6, 31.3, 62.5, and 125 nM of dCTP (Invitrogen), dCMP (Sigma), dUTP (Sigma), and dUMP (Sigma)] were analyzed by UPLC-MS/MS using an Acquity Ultra Performance LC system (Waters) coupled with a Xevo TQ-S mass spectrometer (Waters) with an ESI source in negative ion mode.

## Genomic extraction and quantification using qPCR

Phage genomes were extracted as previously described [56]. Briefly, phage lysates were treated with RNase A (Roche; 1 μg/mL), DNase I (NEB; 18 U), and lysozyme (Sigma-Aldrich; 1 mg/mL). Samples were incubated at 37˚C for 90 min, and then the DNase was inactivated by incubating at 75˚C for 10 min. The samples were then further treated with 0.1 mg/mL Proteinase K (Invitrogen) and 0.5% SDS and were incubated at 55˚C for 1 h. Samples were then extracted once with phenol-chloroform: isoamyl alcohol (25:24:1) and a second time with chloroform. DNA was isolated by ethanol precipitation with the addition of 0.3 M sodium acetate. DNA quality and quantity were determined using a NanoDrop spectrophotometer (Thermo Fischer Scientific).

For measuring phage genome abundance, 25 μL reactions consisted of 5 μL each 0.625 μM primers 1 and 2, 12.5 μL 2X SYBR master mix, and 2.5 μL of 2.5 ng/μL phage genomic DNA. qPCR reactions were performed in technical duplicates for biological triplicate samples. The relative abundance was calculated by comparing the $C_t$ values of phage infected *E. coli* with AvcID to inactive AvcID* at each timepoints.

For measuring phage DNA replication, overnight cultures were subcultured 1:100 in LB at 35˚C and grown to OD600 of ~0.3. The cultures were then infected with T5 phage at the final MOI of 1. Then, 1.5 ml of culture was collected at 10, 20, 30 and 40 min post-infection. Culture aliquots were centrifuged at 15,000g for 1 min, and the pellets were flash frozen in a dry ice–ethanol slurry. DNA was extracted using Wizard Genomic DNA Purification Kit (Promega), using the Gram-negative bacteria protocol and purified DNA from each sample was uniformly resuspended in 50 μl of DNA dehydration solution. DNA quality and quantity were determined using a NanoDrop spectrophotometer. Primers targeting the T7 p52 genes were used to quantify the abundance of T7 phage genome in each sample with qPCR as described above. The relative abundance of T7 genomes was calculated using the difference of Ct between 5 min and each subsequent timepoint (2(−ΔCt)) for each strain.

## Alkylation of T7 phage

Alkylation of T7 phage was performed as previously described with slight modifications [31]. Phage lysates were treated for 2 hr at 37˚C either with 0.01 M methyl methanesulfonate (MMS) (Santa Cruz Biotechnology) or equal volume of phage buffer (10 mM Tris, pH 7.6, 10mM $MgCl_2$). The treated phage was then chilled on ice for 10 minutes and then dialyzed overnight at 4˚C against phage buffer with 10 kDA dialysis tube (Thermo Scientific). Phage titers after alkylation and dialysis were determined using the small-drop plaque assay.

## Transmission electron microscopy (TEM)

High titer phage stocks were prepared using a 15 mL soft agar overlay with 150 μL of *E. coli* DH10B and 150 μL of 2.26x10^10 PFU/mL (T5 or T7). Liquid cultures were then grown using 30 mL LB with 100 μL plus 0.5 mL overnight culture harboring a plasmid with active or inactive *avcID* system (AmpR) and one plaque (either T5 or T7). The culture was grown at RT while shaking at 200 RPM for 6 hours or until clear. 3mL of chloroform was added and the culture was incubated for an additional 5 minutes before being centrifuged at 8,000 x g (~7,000 RPM in F14-14x50cy rotor) for 10 minutes at 4˚C. The supernatant was then spun at 26,000 x g (~12,500 RPM in F14-14x50cy rotor) for 90 minutes at 4˚C to pellet the phage. The pellet was resuspended in 1.5 mL of phage buffer by nutating overnight at 4˚C.

Approximately ~5 μL of phage samples were applied to freshly glow discharged (PELCO easiGlow, 15 mA, 45 s) continuous carbon support film grids (Ted Pella, Cat. No: 1754-F) for 60 seconds, followed by washing with distilled water and then staining with 1% aqueous

Uranyl Acetate (Electron Microscopy Solutions, Cat. No: 22400–4). Grids were blotted dry with Whatman filter paper. The phage samples were imaged at the RTSF Cryo-EM Core Facility at Michigan State University using a Talos Arctica operated at 200 keV. Micrographs were collected on a Ceta camera at a nominal magnification of 45,000 (2.2 Å/pixel) and 57,000 (1.8 Å/pixel) with an exposure time of 1 second and objective lens defocus setting of 5 μM underfocus.

## Statistical analysis

As specified in the figure legends, all of the statistical analyses were analyzed in GraphPad Prism Software. Statistical significances are denoted as follows: a single asterisk (*) indicates $p < 0.05$; double asterisks (**) indicate $p < 0.01$; triple asterisks (***) indicate $p < 0.001$; and quadruple asterisks (****) indicate $p < 0.0001$. Means ± SEM and specific n values are reported in each figure legend. All data is accessible at: Waters, Christopher; Hsueh, Brian (2023). Replication cycle timing determines phage sensitivity to a cytidine deaminase toxin/ antitoxin bacterial defense system. Dataset. https://doi.org/10.6084/m9.figshare.22028480.v1.

## Supporting information

**S1 Fig. The raw images of the Northern blots and Western blots of Fig 1.** Shown are the raw images of Northern blots of *avcI* RNA and Western blots of AvcD-6xHis during rifampicin treatment, spectinomycin treatment, T5 infection, and T7 infection. Gray triangle corresponds to *avcI* transcript (~280 nt) and black triangle corresponds to AvcD-6xHis (60 kDa).
(DOCX)

**S2 Fig. Replicates of TEM Images.** Transmission Electron Microscopy (TEM) of *Escherichia coli* bacteriophage T7 (row a) and T5 (rows b and c) infecting *E. coli* host carrying the active *avcID* (row a and c) or inactive *avcID* (row b). Samples were negative stained with 1% (w/v) uranyl acetate. Scale bar 100 nm.
(DOCX)

**S3 Fig. AvcID shows poor protection against T7 phage infection.** The relative genome abundance of T7 infecting *E. coli* MG1655 pAvcID or inactive pAvcID*. Data represent the mean ± SEM of three biological replicate cultures, two-way ANOVA with two-sided Šídák's multiple-comparison test.
(DOCX)

**S4 Fig. The AvcID system provides modest protection in *E. coli* against MMS-treated T7 phage.** Growth curves for *E. coli* with active (pAvcID) or inactive (pAvcID*) AvcID system after infection with untreated T7 (**A, C**), or MMS-treated T7 phage (**B, D**) at multiplicities of infection (MOI) of 0.1 or 0.0001. Data represents the mean ± SEM of three biological replicate cultures.
(DOCX)

**S5 Fig. Growth curves of *E. coli* infected with T7 mutant phages.** Growth curves of *E. coli* with active (pAvcID) or inactive (pAvcID*) AvcID system after infection with T7 mutants, T7$^{C74}$ (A) or T7$^{412}$ (B) at various MOIs. Data represents the mean ± SEM of three biological replicate cultures.
(DOCX)

**S1 Table. Bacterial strains and phages used in this study.**
(DOCX)

**S2 Table. Plasmids Descriptions.**
(DOCX)

**S3 Table. Oligonucleotides Used in This Study.**
(DOCX)

## Acknowledgments

We thank Ry Young and Ian Molineux for their valuable suggestions and Ian Molineux for providing us with the T7 mutant phages. We thank Sundharraman Subramanian and Kristin Parent for the TEM imagining as well as the MSU RTSF Cryo-EM facility. We thank Keifei Yu for providing us with the NR8052 *E. coli* strain. We also thank MSU RTSF mass spectrometry facility core for their technical support.

## Author Contributions

**Conceptualization:** Brian Y. Hsueh, Christopher M. Waters.

**Data curation:** Brian Y. Hsueh, Micah J. Ferrell, Ram Sanath-Kumar, Amber M. Bedore.

**Formal analysis:** Brian Y. Hsueh, Micah J. Ferrell, Ram Sanath-Kumar, Amber M. Bedore.

**Funding acquisition:** Christopher M. Waters.

**Investigation:** Brian Y. Hsueh.

**Methodology:** Brian Y. Hsueh.

**Project administration:** Christopher M. Waters.

**Supervision:** Christopher M. Waters.

**Writing – original draft:** Brian Y. Hsueh, Christopher M. Waters.

**Writing – review & editing:** Brian Y. Hsueh, Christopher M. Waters.

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
