## [Decision Letter · Decision Letter 0]

16 Mar 2023

Dear Dr. Waters,

Thank you very much for submitting your manuscript "Time to lysis determines phage sensitivity to a cytidine deaminase toxin/antitoxin bacterial defense system" for consideration at PLOS Pathogens. As with all papers reviewed by the journal, your manuscript was reviewed by members of the editorial board and by several independent reviewers. In light of the reviews (below this email), we would like to invite the resubmission of a significantly-revised version that takes into account the reviewers' comments.

As you will see, the two reviewers each appreciated the general topic and message of the paper, but they also each raised significant concerns. In particular, both had concerns about whether lysis timing per se is the key determinant of AvcID effectiveness, and Rev 1 raised the alternative possibility of T7 encoding an inhibitor. There were also concerns raised about some of the key technical aspects of the work, such as the MOIs used and the consequent behavior of cells infected at different MOIs. If the issues raised by the reveiwers can all be addressed, the authors should submit a significantly revised manuscript.

We cannot make any decision about publication until we have seen the revised manuscript and your response to the reviewers' comments. Your revised manuscript is also likely to be sent to reviewers for further evaluation.

Sincerely,

Michael T. Laub

Guest Editor

PLOS Pathogens

Karla Satchell

Section Editor

PLOS Pathogens

Kasturi Haldar

Editor-in-Chief

PLOS Pathogens

orcid.org/0000-0001-5065-158X

Michael Malim

Editor-in-Chief

PLOS Pathogens

orcid.org/0000-0002-7699-2064

As you will see, the two reviewers each appreciated the general topic and message of the paper, but they also each raised significant concerns. In particular, both had concerns about whether lysis timing per se is the key determinant of AvcID effectiveness, and Rev 1 raised the alternative possibility of T7 encoding an inhibitor. There were also concerns raised about some of the key technical aspects of the work, such as the MOIs used and the consequent behavior of cells infected at different MOIs. If the issues raised by the reveiwers can all be addressed, the authors should submit a significantly revised manuscript.

Reviewer's Responses to Questions

**Part I - Summary**

Reviewer #1: Summary of the paper

This work is a continuation of a previous work from the same lab which identified the AvcID TA system, characterized its mechanism of action and its role in phage defense (ref 23 of this work). In this work, the authors are focusing on the triggering mechanism of the AvcID system and its impact on phage propagation. They find that AvcI transcript levels are strongly reduced upon phage infection or upon addition of antibiotics that block transcription, suggesting that blocking of chromosomal transcription by phages leads to rapid loss of AvcI transcript due to its inherently short half-life. Surprisingly, they find that the system is activated (as judged both by degradation of AvcI and by the deamination of dC into dT by AvcD) both by phage T5 and T7, but that only the former is affected by this activity in terms of phage propagation. The authors hypothesize that this is due to the shorter lytic cycle of T7 which mitigate the impact of deamination on phage replication. The authors demonstrate that a T7 mutant with a longer lytic cycle is sensitive to AvcID defense. Monitoring multiple phages suggest a (rather weak) correlation between lytic cycle length and ability to defend. As a side note, the author present data that suggest that the impact of dC deamination does not arise most likely from the increase in the level of dU which suggests it arises from reduction in the level of dC.

General assessment

I think that the data presented here indeed show that AvcI transcript levels are strongly reduced upon cessation of transcription and that this most likely suggests that phage inhibition of chromosomal transcription is the triggering mechanism. I find the data on the activation of the system by both T5 and T7 to be really interesting, but I am not very persuaded that the only explanation is the difference in lysis time (maybe more accurately described as difference in replication time – lysis time is the observed proxy for it). While this is an elegant proposal, I do not think that the data properly entangle the difference in replication/lysis time from other confounding factors which control the replication processes. Specifically, I am not persuaded by either Fig. 7 or Fig. 8 that lysis time is the major parameter controlling this behavior.

As can be seen below, I do not ask for many additional experiments, but think that the authors should be more careful with their interpretation.

Reviewer #2: In the manuscript by Hsueh, et al., the authors investigate AvcID, a newly described TA system that limits phage replication through activation of a cytidine deaminase toxin AvcD. The main focus of this paper is determining why some phages are impeded by AvcID and others are not. The authors put forward the provocative hypothesis, that time to lysis is the driving attribute that defines AvcID defense against T5 vs. T7 phages. While many experiments produce useful results for understanding the AvcID system the conclusions put forward overstate these findings or do not adequately acknowledge alternative hypotheses.

One structural issue the authors are facing is that the difference between T7 and T5 phages include far more than phage lysis timing. Perhaps the most significant of these is differences in expression of defense system inhibiting proteins. A simple explanation for why T7 is not restricted by AvcID and T5 is, could be that T7 encodes an AvcID inhibitor that is expressed only during genome replication, thereby allowing the phage to circumvent dCTP/dCMP depletion (this hypothetical time might occur be before the 15 min timepoint in Fig. 3). The assertion that lysis timing is correlated but not causative is supported by Figure 8, which shows no relation between lysis timing and phage defense.

An additional challenge facing this study is that AvcID from V. parahaemolyticus appears to not provide strong phage defense phenotypes in these assays. In Figure 1, an infection at an MOI of 0.1 collapses the culture similarly to an MOI of 1. At an MOI of 0.1, only 10% of the culture should be infected. If AvcID is protecting the population–why is the remaining 90% of the culture, which is uninfected, unable to grow or sustain an OD600?

To improve the manuscript in its current form, the following specific critiques should be addressed:

**Part II – Major Issues: Key Experiments Required for Acceptance**

Reviewer #1: 1. Figure 8: This figure is supposed to show a correlation between lysis time and the effect of the AvcID system on phage propagation. Here are several comments:

a. This is not very persuasive, as many phages with long cycle time are not effectively inhibited by the system (not only T4, but also lambda(vir) and SECphi27, both of which have a longer lysis time than T7412).

b. In addition, the previous work showed that this version of AvcID (from V. parahemoliticus) strongly defends against phage T3, as is also mentioned in the introduction of this paper. This is not reflected for some reason in the current data. This difference needs to be resolved. T3 has a fairly short lysis time according to figure 8, so the ability of AvcID to defend against it also refutes the proposed mechanism of selectivity between phages.

2. Alternative explanations for the difference between phages. The authors mention one alternative mechanism for the difference between T7 and T5 – that T7 is utilizing the degradation of the bacterial chromosome to enrich its dC pool. I assume that this implies that T5 does not do that? Could it be that the T7 412 mutant is defective in this ability? As far as I remember, chromosomal degradation is guided by early T7 genes, which are located in the same area where T7 412 is mutated. Other alternative explanations should also be discussed.

3. MMS experiment: it is very hard to understand the exact logic of the experiment. Why exactly did the authors used two different MOIs and why is the time delay between the strains with and without the system not the same in the two MOIs? As the time delay depends only on the sensitivity to defense of the first round of replication, I would assume it should be the same irrespective of the number of following rounds, no?

4. The discussion of the Dut and Ung results is a bit off the main line of argument of this paper. I understand that the authors want to share thise piece of data and do not want to make a whole paper out of it, so they add it here. Yet, maybe they could improve the way they introduce it to the narrative of this work.

Reviewer #2: Major critiques:

1. Based in Fig. 1, the AvcID system only defends against phage T5 at very low MOIs (MOI≤0.001). When higher MOIs are tested, E. coli cultures expressing the active avcID system still collapse upon phage infection. Further, there does not seem to be a difference in the timing of culture collapse between different MOIs of 1, 0.1, and 0.01. The authors should: (a) plot each of the graphs in Fig. 1 on the same y axis to aid interpretation, (b) explain why the culture collapse kinetics are not different by 10-fold between MOIs of 1, 0.1, and 0.01.

2. In Fig. 2, the authors convincingly show that inhibition of transcription leads to decreased AvcI without affecting levels of AvcD. They further show that infection with phage T5 and T7 also leads to a decrease in AvcI. However, the conclusion that T5 and T7 are decreasing AvcI via host transcription shutoff (line 180 and elsewhere) is not fully justified. Although host shutoff has been reported for both phages, these phage also rapidly destroy the host chromosome during initial steps of infection. (a) The text should be clarified to accommodate that changes in avcI levels are not necessarily the result of RNA polymerase inhibition or Southern blot analysis should be used to show the avcI is still present. (b) It is essential that the authors include loading controls (e.g. house keeping genes) for their northern and western blots in this figure to aid interpretation.

3. Figure 3 presents a central issue with the manuscript. If AvcID results in a decrease in CTP/CMP in phage T7 similar to the decrease in phage T5, why is there no phage defense? I can envision two hypotheses: either genome replication has occurred prior to 15 minutes when nucleotides are measured or T7 genome replication is unaffected by decreased CTP/CMP. These data do not support a role of lysis timing as lysis has not been initiated but AvcID is clearly active. Yes, T7412 may argue that extending lysis timing may enhance susceptibility to AvcID, but T7412 also has numerous other issues with gene expression that could result in decreased inhibitor expression or changes in other aspects of phage physiology (as evidenced by the smaller plaques in Fig. 7). To substantiate their current conclusions, the authors should clarify how T7 lysis timing, which is still after AvcID activation, results in no decrease in PFU production. Alternatively, T7 genome replication and CTP/CMP should be measured at earlier timepoints. These experiments may reveal that it is not lysis timing, but rather genome replication timing that is the important driver in AvcID susceptibility.

4. For Figure 8, why is there no ∆AUC for phage T3? This phage was among those which was inhibited in a previous report (line 129). Do these findings suggest that a liquid assay is indeed not more robust than a double agar overlay assay (line 137)?

5. The characterization of “defective” phages should be improved and clarified. TEM images of defective phages are convincing (and interesting!) but inherently qualitative. Can the authors improve their analysis through quantitative analysis of TEM images?

**Part III – Minor Issues: Editorial and Data Presentation Modifications**

Reviewer #1: 5. CFU measurements of T5 (paragraph starting on line 185, Fig. 4A): The protocol here is unclear. Was infection done at low MOI? In this case the data in Fig. 4A should mean that with the system a collapse of the population is still observed, but is delayed by ~60 minutes compared to without the system (or with a mutated one). This goes in line with the partial effect of the system on T5 propagation that is seen by other means in this section. I think that saying that population collapse is delayed is more accurate than simply saying that there is a big difference at the one time point where there is a big difference (i.e 2 hrs, line 191).

6. dC pool depletion (line 347): from the experiment using the Ung deletion, the authors suggest that the problem imposed by AvcD to the phage is the depletion of the dC pool and not the increase of the dU pool. One way to study this is to add to the medium dC (dCMP or dCTP) and monitor whether this rescues T5 infection. This may not work if export of this molecule is not sufficiently high, but worth a try.

7. Is AvcID an abortive infection system (paragraph starting in line 375, Fig. 1)? The authors discuss whether AvcID is an abortive infection system. Here are some comments on that.

a. As the overexpression of AvcD does not lead to cell death, we would not really expect a priory that its release from AvcI would lead to this effect. In this respect, the system is not abortive simply because the “toxin” is not toxic.

b. I think that the results shown in Fig. 1 cannot support or negate an “abortive” behavior, simply because it seems that AvcID only provides weak defense against T5 and does not block lysis by the phage (and actual release of functional virions). Under these conditions, it is very hard to say anything directly on the effect of the system of the cells decoupled from the effect of the phage.

c. It is worth noting that an MOI of 1 is anyway insufficient to easily determine an abortive infection effect as ~30% of the cells are not infected initially.

8. AvcI degradation (Figure 2 and S1, S2):

a. One would assume that all the perturbations done by the authors in Fig. 2 should control the production of the AvcI RNA and not its degradation dynamics. It is therefore inappropriate to ascribe the difference in the decay time of the RNA under different conditions to changes in degradation. It may depend maybe on changes in the dynamics of expression inhibition. I don’t have any issue with the results, just a bit more care is needed with their interpretation.

b. There are two bands in the northern blot of AvcI, as clearly seen in the T5 data – why is that? Are they the same? Do the authors quantify both of them or only one of them?

c. Quantification (in S2) is probably very rough. It is very clear from the images in S1 and Fig 2 (note confusion in figure references in the supp. Text, line 5) that the T5 bands in time 0 are overexposed. As this leads to overestimation of the degradation time and not to underestimation, I do not think it is very important, but it should be noted in the text when discussing the degradation time.

9. Confusing choice of words. Consider rephrasing:

a. line 101 – “Similarly, type I-IV TA systems have demonstrated that one of their primary physiological roles is to limit phage infections”

b. line 105 - some words missing here – “They also are highly abundant in free-living bacteria but not symbiotic, host-associated species [22], supporting that MGEs are evolutionarily beneficial and important in bacteria that are constantly challenged by phages.”

c. Line 137: “protection in liquid culture as this would be a more robust experimental system to explore molecular mechanism.”

Reviewer #2: Minor critiques:

1. The authors should provide an explanation for why two bands appear for avcI in their northern blot, especially as these seem inconsistent.

2. The growth curves in Fig. 1 should show OD and growth before infection, then indicate the timepoint in which phage was added (or indicate that x-axis is minutes post-infection).

3. Can the authors clarify why T5 phage defense data (MOI=0.1) in Fig. 1 is inconsistent with phage defense data in Fig. 6A?

4. In Fig. 2D, the authors state that protein stability of AvcD is not affected. However, AvcD-6xHis is degraded after 20 minutes during infection with T7. Is this significant? Is this, perhaps, the important facet of why T7 is not impacted by AvcD?

5. In Fig. 4, infected bacterial pellets are separated from phage lysate before quantification of CFU/mL of the bacteria and PFU/mL of the phage. Is there a way to control for phage that have adsorbed onto the bacterial culture?

6. Labels above TEM images would increase clarity and ease of comprehension (Fig. 5).

7. T5 encodes a protein that may influence the activity of Ung (PMCID: PMC8201957). A discussion of this in relation to data in Fig. 6 would be helpful and could perhaps explain the lack of phenotype.

8. Figure 8: These data could be improved by inclusion of replicates, error bars, etc.

PLOS authors have the option to publish the peer review history of their article (what does this mean?). If published, this will include your full peer review and any attached files.

Reviewer #1: No

Reviewer #2: No
---

## [Decision Letter · Decision Letter 1]

17 Jul 2023

Dear Dr. Waters,

Thank you very much for submitting your manuscript "Replication cycle timing determines phage sensitivity to a cytidine deaminase toxin/antitoxin bacterial defense system" for consideration at PLOS Pathogens. As with all papers reviewed by the journal, your manuscript was reviewed by members of the editorial board and by several independent reviewers. The reviewers appreciated the attention to an important topic. Based on the reviews, we are likely to accept this manuscript for publication, providing that you modify the manuscript according to the review recommendations.

The reviewers are both generally satisfied with the revisions made and the paper can be formally accepted once the authors deal with the very minor remaining comments from Reviewer 1.

Sincerely,

Michael T. Laub

Guest Editor

PLOS Pathogens

Karla Satchell

Section Editor

PLOS Pathogens

Kasturi Haldar

Editor-in-Chief

PLOS Pathogens

orcid.org/0000-0001-5065-158X

Michael Malim

Editor-in-Chief

PLOS Pathogens

orcid.org/0000-0002-7699-2064

The reviewers are both generally satisfied with the revisions made and the paper can be formally accepted once the authors deal with the very minor remaining comments from Reviewer 1.

Reviewer Comments (if any, and for reference):

Reviewer's Responses to Questions

**Part I - Summary**

Reviewer #1: I am generally pleased with the revision of the paper. The reviewers both added experiments that support their main claim and eliminate or soften claims that were not sufficiently substantiated.

I approve the acceptance of this manuscript with minor revisions.

Reviewer #2: The authors have satisfied all of my previous concerns

**Part II – Major Issues: Key Experiments Required for Acceptance**

Reviewer #1: (No Response)

Reviewer #2: (No Response)

**Part III – Minor Issues: Editorial and Data Presentation Modifications**

Reviewer #1: 1. Fig. 1B – is T7 replication time so fast and burst size so high that MOI of 0.0001 sufficient to collapse it in 100 mins? If so, why don’t we see an earlier collapse with higher MOIs?

2. Line 193 – which MOI is used in this experiment? I guess this is a high MOI, otherwise this result contradict Fig. 1A?

3. Line 108 – typo in “deoxyctidine”

Reviewer #2: (No Response)

PLOS authors have the option to publish the peer review history of their article (what does this mean?). If published, this will include your full peer review and any attached files.

Reviewer #1: No

Reviewer #2: No

Figure Files:

Data Requirements:

Reproducibility:

References:

---

## [Editor Report · Decision Letter 2]

21 Jul 2023

Dear Dr. Waters,

We are pleased to inform you that your manuscript 'Replication cycle timing determines phage sensitivity to a cytidine deaminase toxin/antitoxin bacterial defense system' has been provisionally accepted for publication in PLOS Pathogens.

Best regards,

Michael T. Laub

Guest Editor

PLOS Pathogens

Karla Satchell

Section Editor

PLOS Pathogens

Kasturi Haldar

Editor-in-Chief

PLOS Pathogens

orcid.org/0000-0001-5065-158X

Michael Malim

Editor-in-Chief

PLOS Pathogens

orcid.org/0000-0002-7699-2064
---

## [Editor Report · Acceptance letter]

6 Sep 2023

Dear Dr. Waters,

We are delighted to inform you that your manuscript, "Replication cycle timing determines phage sensitivity to a cytidine deaminase toxin/antitoxin bacterial defense system," has been formally accepted for publication in PLOS Pathogens.

Best regards,

Kasturi Haldar

Editor-in-Chief

PLOS Pathogens

orcid.org/0000-0001-5065-158X

Michael Malim

Editor-in-Chief

PLOS Pathogens

orcid.org/0000-0002-7699-2064